# Long-Term Performance Evolution of RIOHTrack Pavement Surface Layer Based on DMA Method

**DOI:** 10.3390/ma15186461

**Published:** 2022-09-17

**Authors:** Zhimin Ma, Xudong Wang, Yanzhu Wang, Xingye Zhou, Yang Wu

**Affiliations:** 1Research Institute of Highway Ministry of Transport, Beijing 100088, China; 2National Observation and Research Station of Corrosion of Road Materials and Engineering Safety in Dadushe Beijing, Research Institute of Highway Ministry of Transport, Beijing 100088, China; 3School of Transportation Science and Engineering, Harbin Institute of Technology, Harbin 150090, China

**Keywords:** asphalt mixture, phase transformation, characteristic temperature, dynamic mechanical analysis, load, environment

## Abstract

Asphalt mixture is a typical viscoelastic material, and its road performance will change with the action of environment and load during actual service. This study conducted experimental research on the surface course asphalt mixture of three categories and six typical structures of RIOHTrack based on the Dynamic Mechanical Analysis method. Moreover, this study explored the performance evolution law of asphalt mixture under the coupling action of load and environment in the process of loading from 0 million to 54 million standard axle times. Results demonstrated that the phase transition characteristic temperature of the surface course materials of the three types of typical structures showed a trend of first increasing and then decreasing with the accumulation of load and environmental effects, indicating the presence of two stages of the dual coupling effect of environmental aging and load rolling on the asphalt mixture during service. In addition, the results suggested that the phase transition characteristic temperature, modulus, and phase angle of the surface layer materials have obvious material differences and structure dependencies.

## 1. Introduction

Asphalt pavement in the whole life cycle of service process degrades by the environment and vehicle load coupling. The service performance of each structural layer shows a gradual decay until the state of destruction. The surface condition, skid resistance, and structural strength of the pavement are constantly attenuated with the continuous action of the load and the environment, resulting in a gradual decline in the bearing capacity of the pavement and eventually failure and damage. Some asphalt pavements have exceeded their design life without structural damage in the actual service process. Meanwhile, other pavements have been damaged before reaching the design lifetime [1].

The construction of more durable long-life asphalt pavements has been an important development direction in the field of road engineering. The construction of long-life asphalt pavement construction and maintenance technology system includes several aspects of research: (1) research on the damage mechanism of pavement materials and balanced design methods; (2) research on the structural design methods and indicators of long-life asphalt pavement; (3) research on the integration of asphalt pavement design and construction technology. The evaluation of the service performance of in-service pavements and the study of the mechanical state damage evolution law have been the main bottleneck that restricts the construction of long-life asphalt pavement design methods. The performance evolution law of asphalt pavement in service is very complex, subject to both environmental and loading effects, and at the same time closely related to the physical and mechanical properties of each structural layer material. The same material under different temperatures and loads exhibits very different mechanical response states, which in turn makes the overall service state of asphalt pavements vary significantly. Therefore, understanding how to evaluate the bearing capacity and service status of the active pavement and how to reveal the damage evolution law of pavement materials and structures is an important prerequisite and theoretical foundation for asphalt pavement to achieve long-life goals.

In order to achieve the evaluation of in-service pavements in service condition, researchers have conducted a large number of tests on asphalt pavements for a long time. These tests are roughly divided into two categories. The first category is outdoor tests represented by Benkelman beam deflectometers, drop-weight deflectometers, and load-bearing plate tests, which are often used to evaluate the bearing capacity of pavement structures on site. Refs. [2,3,4,5] used a portable falling weight deflectometer (PFWD) to study the relationship between dynamic elastic modulus and compactness and water content. They found that the correlation coefficient between dynamic elastic modulus and static elastic modulus, compactness, and water content was greater than 0.85, and the maximum value of dynamic elastic modulus was close to the optimal water content, which is consistent with the actual situation. The results showed that PFWD can be used for the rapid detection and evaluation of subgrade resilience. Zha Xudong [6,7] determined the data analysis method of PFWD rapid detection of subgrade modulus by analyzing the dynamic test principle of portable drop weight deflector and established the correlation between PFWD modulus and bearing plate resilience modulus, Beckmann beam deflection, and FWD modulus through a large number of comparative tests of subgrade on site. The results showed that PFWD had good correlation with other detection methods. However, this type of test method represents the bearing capacity of the overall structure from the subgrade, the base layer to the surface layer, and cannot reflect its real bearing capacity for a specific layer [8]. The second category is to conduct laboratory tests on asphalt mixtures obtained by drilling cores in the field and characterize the service performance of specific layers through the test results of the materials. The most common mechanism is to use UTM and MTS to apply sinusoidal load to the material and measure its dynamic modulus, loss modulus, and phase angle to evaluate the mechanical properties of the material. However, the limitation of this method is that the required size of the test piece is particularly large. When evaluating the service state of the active pavement, the damage to the pavement is large, which is not conducive to long-term testing. Neither of these two types of methods can achieve a long-term study of the service condition of in-service pavements due to the limitations mentioned above.

Dynamic mechanical analysis as an important means of testing the properties of viscoelastic polymer materials, not only the required size of the specimen is small, but also can simulate a variety of loading and temperature environments, so the method is more widely used in the field of polymer materials, especially composite materials. Asphalt mixture is a typical viscoelastic material that exhibits distinct mechanical properties under various external conditions and viscoelastic properties under different temperatures and loading modes. Asphalt mixtures can be considered to be close to elastic under high-speed vehicle loads even in summer, while slow temperature shrinkage exhibits viscous flow properties in winter [9]. The phase state of the asphalt mixture changed from a glassy state to a highly elastic state with the gradual increase in temperature, eventually reaching a viscous flow state. Relevant studies have shown that the cracking, rutting, fatigue, and other damage problems of asphalt pavement are closely related to the phase transition and viscoelasticity of asphalt mixtures [10]. Therefore, the study of the phase state of the asphalt mixture is important for characterizing the service state of the pavement. Therefore, in this study, the test method of dynamic mechanical analysis was chosen to investigate the service condition of in-service pavement materials in relation to the service condition [11].

For the reasons stated above, this study relies on the Research Institute of Highway Ministry of transport, RIOHTrack, to carry out pavement coring for the surface layer of the three types of track structures from 0 loading, with 6 and 8 million times of standard axle load as the cycles. A continuous temperature sweep test based on a fixed frequency and a heating rate was carried out using the DMA slice test method, and the change law of the characteristic temperature of the phase transition of the asphalt mixture with the number of loads and the material and structural differences of the characteristic temperature were analyzed.

## 2. Materials and Tests

### 2.1. Materials

#### 2.1.1. Road Coring

This research is based on the first full-scale track in the field of road engineering in China, the Research Institute of Highway Ministry of transport (RIOHTrack), which is located in the road traffic test site of the Ministry of Transport, Majuqiao, Tongzhou District, Beijing. The RIOHTrack test road is an elliptical closed curve composed of a straight line and a circular curve. This road runs north–south and is symmetrically arranged, with a total length of 2039 m. According to the linear characteristics of the test road and the purpose of the test, RIOHTrack has paved a total of 38 pavement structures, of which 19 types of asphalt pavement structures are found in seven categories. In this study, six typical structures of three categories were selected for research.

RIOHTrack has been loaded since November 2016. According to the load level, the corresponding action times converted to standard axle load (0.7 MPa) have been loaded for more than 60 million times, which is equivalent to the load level of China’s heavy haul traffic for 30 years. This study took 6 and 8 million times of standard axle loads as the cycles to carry out pavement coring to study the service performance evolution of asphalt mixture in the whole process under the coupling effect of environmental load. The coring information is illustrated in Table 1, and the obtained core samples (part) are shown in Figure 1.

#### 2.1.2. Material Parameters

This study focuses on RIOHTrack’s three major categories of six types of pavement structure surface layers, namely, semi-rigid base asphalt surface layer (STR1, 3), flip-chip structure (STR10, 12), and thick asphalt surface layer structure (STR18, 19). The surface layer material of STR1/3 and 10/12 is AC-13, and that of STR18/19 is SMA-13. The asphalt used in both materials is SBS modified asphalt. The schematic of each structure and the basic technical parameters of the two materials are shown in Figure 2 and Table 2 and Table 3.

### 2.2. Test

#### 2.2.1. Specimen Preparation

The core samples were drilled on site as cylinders with a diameter of 15 cm. In this study, the dynamic mechanical test method was used, and the test piece was a sheet-like body with a 60 mm length, 15 mm width, and 3.5 mm thickness, as shown in Figure 3. The cutting method of the test piece is as follows: first, the core sample is cut according to the structure of each layer of the asphalt pavement to obtain the cylindrical core sample of the upper layer, the middle surface layer, and the lower layer structure; a precision cutting machine is then used to cut it into the test piece required for the DMA test. The process is shown in Figure 4 [12,13,14].

#### 2.2.2. Test Principle

The dynamic mechanical behavior of a material refers to the strain (or stress) response of the material under the action of alternating stress (or strain). The most commonly used alternating stress in the dynamic mechanical test method is the sinusoidal stress, which can be expressed as follows:(1)$$\tau \left(t\right)=\tau_{0}sin\omega t$$
where τ_0_ is the stress amplitude, *ω* is the angular frequency (unit: radians), and the strain response of the specimen under sinusoidal alternating stress varies with the material properties.

Asphalt materials are viscoelastic materials, and the strain will lag behind the stress by a phase angle *δ* (0° < *δ* < 90°).
(2)$$\gamma \left(t\right)=\gamma_{0}\sin\left(\omega t-\delta \right)$$

Expanding Formula (2) yields
(3)$$\gamma t=\gamma_{0}\left(cos\delta sin\omega t-sin\delta cos\omega t\right)$$

The strain response includes two items: the first term is in phase with the stress, reflecting the elasticity of the material; the second term is 90° behind the stress, reflecting the viscosity of the material.

If a sinusoidal strain is applied to the viscoelastic specimen:(4)$$\gamma \left(t\right)=\gamma_{0}\sin\left(\omega t\right)$$
then the stress response of the specimen will lead the strain by a phase angle *δ*:(5)$$\tau \left(t\right)=\tau_{0}sin\omega \left(\omega t+\delta \right)$$

The modulus of a material is the ratio of stress to strain, and the resulting modulus should be a complex number due to the phase difference between stress and strain in viscoelastic materials. The stress–strain function is written in complex form for convenience of calculation:(6)$$\gamma \left(t\right)=\gamma_{0}\exp\left(i\omega t\right)$$
and
(7)$$\tau t=\tau_{0}\exp\left[i\left(\omega t+\delta \right)\right]$$

Thus, complex modulus *E** is:(8)$$E^{*}=\frac{\tau \left(t\right)}{\gamma \left(t\right)}=\frac{\tau_{0}}{\gamma_{0}}e^{i\delta }=\frac{\tau_{0}}{\gamma_{0}}\left(cos\delta +isin\delta \right)$$
that is,
(9)$$E^{*}=\left|E^{*}\right|\left(cos\delta +isin\delta \right)=E^{\prime }+iE^{\prime \prime }$$
where
(10)$$E\prime =\left|E^{*}\right|\cos\delta =\frac{\tau_{0}}{\gamma_{0}}cos\delta$$
(11)$$E\prime \prime =\left|E^{*}\right|\sin\delta =\frac{\tau_{0}}{\gamma_{0}}sin\delta$$
(12)$$E^{*}=\sqrt{E^{\prime 2}+E^{\prime \prime 2}}$$

In the formula, the real number of complex modulus E′ represents the energy stored by the material due to elastic deformation during the deformation, which is called the storage modulus. Imaginary number E″ characterizes the energy lost in the form of heat due to viscous deformation during the deformation of the material, which is called dissipation energy [15,16,17,18].

#### 2.2.3. Experimental Equipment and Methods

In this study, the double cantilever fixture of the dynamic mechanical analyzer (DMAQ800) produced by TA Company (Boston, MA, USA) was used to conduct the test, as shown in Figure 5. Asphalt mixture is a temperature (frequency) sensitive material. Accordingly, the test conditions have a significant effect on the results. After conducting several tests, the research group determined the standard test conditions for the phase temperature of asphalt mixture based on DMA: the strain level was 50 με, the loading frequency was 1 Hz, the temperature sweep range was −30 °C–70 °C, and the heating rate was 2 °C/min.

To ensure the reliability of the test, a force measuring wrench was used to apply the same tightening force, and the test piece cut was fixed to the fixture in accordance with Section 2.2.1. The furnace was closed, the strain and frequency were set, the nitrogen temperature was controlled to −30 °C and kept at a constant temperature for 20 min, and a dynamic sine wave flexural tensile load was applied to the middle of the sliced specimen through the loading axis to carry out the corresponding tests. Five parallel specimens were set for each group of tests.

#### 2.2.4. Analysis Method

The complex modulus *E** of the asphalt mixture specimen, the loss modulus E″, and the tangent value of the phase angle tan φ can be obtained according to the above-mentioned test steps. After the outliers were removed, the data of the parallel specimens were averaged, and the obtained complex modulus curve was inverse S shape. According to the characteristics of the curve, the Boltzmann function (Formula (13)) was used for fitting, the temperature corresponding to the inflection point was taken as the characteristic temperature of phase transition T_1_, and the tangent line was taken through this point. Moreover, the temperatures corresponding to the intersection of the two progressive lines were characteristic temperatures T_2_ and T_3_, as shown in Figure 6.
(13)$$y=\frac{A_{1}-A_{2}}{1+e^{\left(x-x_{0}\right)/dx}}+A_{2}$$
where *A*_1_, *A*_2_, *x*_0_, and *dx* are parameters.

The loss modulus curve obtained from the test had a peak point, and the curves on both sides of the peak point were asymmetrical. Accordingly, the Bigaussian function (Formula (14)) was used for fitting, and the temperature value corresponding to the peak point was taken as T_4_, as shown in Figure 7. The positive curve of the phase angle had a peak point, and the curves on both sides of the peak point were symmetrical. The Gauss function (Formula (15)) was used for fitting, and the temperature value corresponding to the peak point was taken as T_5_, as shown in Figure 8.
(14)$$y=y_{0}+He^{-0.5{\left(\frac{x-xc}{\omega_{1}}\right)}^{2}}\left(x<x_{c}\right);y=y_{0}+He^{-0.5{\left(\frac{x-xc}{\omega_{2}}\right)}^{2}}\left(x>x_{c}\right)\;$$
(15)$$y=y_{0}+\frac{A}{\sqrt[{\omega }]{\pi ∕2}}e^{-2\frac{{\left(x-x_{c}\right)}^{2}}{\omega^{2}}}\;$$
where *y*_0_, *H*, *x_c_*, *ω*_1_, and *ω*_2_ are parameters.

Asphalt mixture is a typical viscoelastic material whose pavement properties change with load and temperature. In this study, the core samples of RIOHTrack with different loading cycles were drilled, and the surface layers of three types and six structures were tested by using the test method described in Section 2.2.3. Five characteristic temperatures T_1_–T_5_ were obtained by using the above-mentioned data processing and analysis methods. T_2_, T_4_, and T_5_ were used to characterize the low, medium, and high temperature properties of the asphalt mixture, respectively. The long-term performance evolution law of the asphalt pavement surface layer under the influence of environment and load was characterized by analyzing the change trend of the characteristic temperature with the increase in load times [19].

## 3. Results and Discussion

### 3.1. Effect of Load-Environment Coupling on the Phase Characteristic Temperature of Asphalt Mixtures

Under the long-term action of environment and load, the service performance of asphalt mixture will change accordingly. The T_2_, T_4_ and T_5_, results of the three major types of structures are calculated and plotted as shown in Figure 9, Figure 10 and Figure 11 to analyze the long-term performance evolution law of asphalt mixture.

T_2_ is the characteristic temperature to characterize the low temperature performance of asphalt mixture, as shown in Figure 9, Figure 10 and Figure 11. With the accumulation of load action, T_2_ of all three types of structures showed a trend of increasing and then decreasing and reached the peak in the range of 30–40 million load actions. This indicates that, in the surface layer of asphalt mixture in the coupling of load and environment, the material from the glassy state into the high elasticity of the degree of difficulty first increased and then decreased, that is, the low temperature performance of the asphalt mixture first deteriorated and then recovered. Analysis of the possible causes make the author believe that: At the beginning of the phase, the effect of the environment is weaker than the effect of the load action, and the pavement material gradually develops fatigue damage, making the viscous specific gravity of the asphalt mixture increase, which is manifested by an increase in the characteristic temperature of the phase transformation. As the load and the environment continue to act, the influence of the environment on the material gradually increases, and the material generates a large number of unsaturated bonds due to aging, which further leads to the occurrence of polymerization reactions, resulting in an increase in the molecular weight of the material, slowing down the movement of the molecular chain, which is manifested as a decrease in the characteristic temperature of the phase transition, and the performance of the pavement in terms of low temperature performance [20].

T_4_ is the characteristic temperature to characterize the medium temperature crack resistance of asphalt mixture. The higher the temperature, the harder the material is, and the more likely it is to produce fatigue cracks. As shown in Figure 9, Figure 10 and Figure 11, T_4_ also shows a trend of increasing and then decreasing under load. This indicates that, during long-term service, there is also a dynamic change process of the medium-temperature crack resistance of the surface layer that is first better and then worse.

T_5_ is the peak temperature of the phase angle tangent. This temperature is the maximum viscous ratio of the material, used to characterize the high temperature performance of the asphalt mixture. The higher the T_5_ temperature, the higher the temperature required to achieve the maximum viscous and elastic ratio of the asphalt mixture at high temperature, that is, the better the high temperature performance of the material. From Figure 9, Figure 10 and Figure 11 can be seen, T_5_ with the cumulative effect of the load, showing a trend of increasing and then decreasing, and T_2_, T_4_ the same evolutionary law, reflecting a good correlation. This indicates that the temperature required to reach the maximum viscous ratio of asphalt mixture first increases and then decreases, in terms of road performance in high temperature service to produce permanent deformation of the degree of difficulty first increases and then decreases, that is, its high temperature performance first becomes better and then worse.

### 3.2. Long-Term Evolution of Modulus and Phase Angle of Asphalt Mixtures

From the analysis in Section 3.1, it can be seen that, in the asphalt mixture in the long-term service process, subject to the cumulative effect of the environment and load, its low-temperature crack resistance first becomes worse and then recovers, while the high-temperature rutting resistance first becomes better and then gradually deteriorates, showing a contradictory relationship, which is consistent with our knowledge. On the other hand, this also reflects that the performance of asphalt pavement during service does not show a constant decay until destruction, but a dynamic change due to the different main influencing factors in the early and late stages, with an inflection point of performance evolution in the middle of service. In order to describe this phenomenon more intuitively, it is necessary to analyze the dynamic modulus and phase angle changes of the asphalt mixture during the whole process of service. Due to the differences in characteristic temperatures, it is not reasonable to select the modulus values corresponding to the characteristic temperatures for comparison, so the complex modulus values corresponding to −20 °C, 20 °C, 60 °C and the phase angle tangent values are selected for analysis in this study, and the results are shown in Figure 12, Figure 13 and Figure 14.

As shown in Figure 12, Figure 13 and Figure 14, under the continuous action of load and environment, both in the low and medium temperature zone and high temperature zone, the complex modulus of asphalt mixture shows a trend of increasing and then fluctuating decrease, indicating that the surface layer of asphalt mixture in service is subjected to environmental aging and the double coupling effect of load crushing. Aging hardens the material. In load rolling there are two stages: when the rutting is small, in the state of compression density is dominant, the material hardens; when the rutting is larger, the mixture produces “flow” deformation. The material becomes soft, which is also a fatigue damage performance. In the long-term service process, the phase angle tangent value of the surface layer of asphalt mixture shows a trend of first decreasing and then increasing, indicating that the asphalt mixture in the early service of the elastic proportion is increasing and viscous proportion is decreasing. In about 30 million loadings the elastic proportion reaches the last stage, after which the elastic part of the proportion is decreasing and viscous proportion is increasing. These phenomena are consistent with the conclusions obtained in Section 3.1. Therefore, it can be considered that, for the mixture in the service process, its performance evolution exhibits two stages: one is the hardening stage, the second is the fluctuation of the soft stage, which for the study of the surface layer of asphalt mixture in service during the mechanical decay law has a certain guiding significance.

### 3.3. Material Variability of Characteristic Temperatures

From the analysis of Section 3.2, it can be seen that the surface layers of the three types of structures show similar evolutionary laws under the action of environmental loading, but there are some differences among them because the surface layer materials of the three types of structures are different. The phase transition characteristic temperatures T_2_, T_4_, and T_5_ of the three types of structures are obtained by fitting and calculating according to the aforementioned experimental analysis method, respectively, and they are plotted as scatter plots and the corresponding trend lines are added, and the results are shown in Figure 15, Figure 16 and Figure 17.

As shown in Figure 15, T_2_ is the characteristic temperature to characterize the low temperature performance of asphalt mixture. The results show that under low temperature conditions, whether it is used for STR1/3 and STR10/12 surface layer of AC-13 or for STR18/19 SMA-13, the trend of the characteristic temperature T_2_ is very close. The author believes that this is because in the low temperature zone, the two asphalt mixtures are in the glassy state, and the material has a high modulus. In the mechanical properties of the performance of the elasticity of the main, i.e., the long-term role of the load, the two performance evolution laws do not show significant differences.

As shown in Figure 16, T_4_ is the characteristic temperature to characterize the medium temperature performance of the asphalt mixture, and the results show that AC-13 and SMA-13 exhibit very different long-term performance evolution patterns in the medium temperature region. The characteristic temperature of AC-13 is lower than that of SMA-13 at the beginning of the loading period, while the characteristic temperature of AC-13 tends to increase with the increase of loading times, while SMA-13 also increases, but its growth rate is much less than that of AC-13. After reaching the peak, both of them show a decreasing trend, and AC-13 also shows a faster decline. This indicates that SMA-13 maintains a more stable medium-temperature performance during the long-term service cycle, while AC-13 is more influenced by the environmental loading. Analyzing the reason is that SMA-13 coarse aggregate content is high, in the mix, the particle surface and surface direct contact, mutual embeddings constitute the skeleton, and improve the temperature sensitivity of the asphalt mixture. At the same time, the higher content of mineral powder and asphalt formation of the material has a high cohesion, to ensure the stability of its overall mechanical properties, so compared with AC-13, SMA-13 by environmental loading. Therefore, SMA-13 performs more consistently than AC-13 over the long-term service cycle under the coupling effect of the loading environment.

As shown in Figure 17, in the high temperature region, SMA-13 has a higher characteristic temperature T_5_ than AC-13 throughout the service cycle and maintains a relatively stable performance. This indicates that the high temperature performance of SMA-13 is always better than that of AC-13 during the full service cycle of asphalt pavement and is less affected by the cumulative effect of environmental load coupling. In addition to the above-mentioned coarse aggregate skeleton embedding, mineral powder and asphalt, the increase of asphalt dosage, and the addition of fibers also played a role in improving the high temperature stability of SMA-13.

### 3.4. Structural Dependence of the Characteristic Temperature

As described in Section 3.2, it can be seen that the trend of the phase transition characteristic temperature is basically the same for STR1/3 and STR10/12 throughout the service cycle because they use the same surface layer material, but there are some differences in the values due to the differences in the pavement structure. In order to analyze the differences, the differences between the three characteristic temperatures of STR1/3 and STR10/12 were calculated as shown in Table 4, Table 5 and Table 6.

As seen in Table 4, the characteristic temperature T_2_ of STR1/3 is about 1–2 °C lower than that of STR10/12 in the pre-service period. While in the middle of service, i.e., when the characteristic temperature reaches its peak, the difference increases with a difference of about 3 °C, and in the late service period, the difference resumes with about 1 °C. This indicates that the structural form of STR1/3 has some advantages in reducing the low temperature cracking performance of the asphalt pavement surface layer.

As shown in Table 5 and Table 6, the characteristic temperature of the thin asphalt surface structure (STR1/3) is higher than that of the thick asphalt surface structure (STR10/12) during the whole service life in the medium and high temperature zone, excluding individual loading stages, which indicates that the thin asphalt surface structure (STR1/3) performs better in terms of high temperature rutting resistance. Combined with the above analysis, it can be judged that STR1/3 has better performance than STR10/12 in both low temperature cracking and high temperature rutting resistance, and the cost of STR1/3 is much lower than STR10/12.

## 4. Conclusions

In this study, the DMA test method was used to test the surface layer asphalt mixes of six asphalt pavement structures in three major categories of RIOHTrack, and the complex modulus, loss modulus, and phase angle of the materials at different loading cycles were obtained and the five-phase transformation characteristic temperatures were obtained by fitting the corresponding functions. The performance evolution law of the in-service pavement surface layer asphalt mixture for the whole process of 0–54 million standard axial loadings was obtained by analyzing the test results, which are summarized as follows.

(1)Under the coupling effect of loading environment, the five-phase transformation characteristic temperatures of the surface layer asphalt mixture all show a trend of first increasing and then decreasing, indicating that with the increase of service time of the pavement, the low temperature crack resistance of the surface layer asphalt mixture first becomes better and then worse, while the high temperature performance first becomes worse and then recovers.(2)With the coupling of load environment, the modulus of asphalt pavement surface layer asphalt mixture first increases and then gradually decreases, while the phase angle first decreases and then gradually increases, indicating that the asphalt mixture in the actual service process, there are two stages of compression density hardening and then fluctuating softness.(3)During the long-term service, the phase transition characteristics of the surface layer asphalt mixture temperature shows obvious material variability and structural dependence, SMA-13 shows more stability than AC-13, and is less affected by the environment and loading, while in terms of structure, the semi-rigid base layer thin asphalt surface structure represented by STR1/3 shows better performance than the inverted structure represented by STR10/12. better road performance.

## Figures and Tables

**Figure 1 materials-15-06461-f001:**
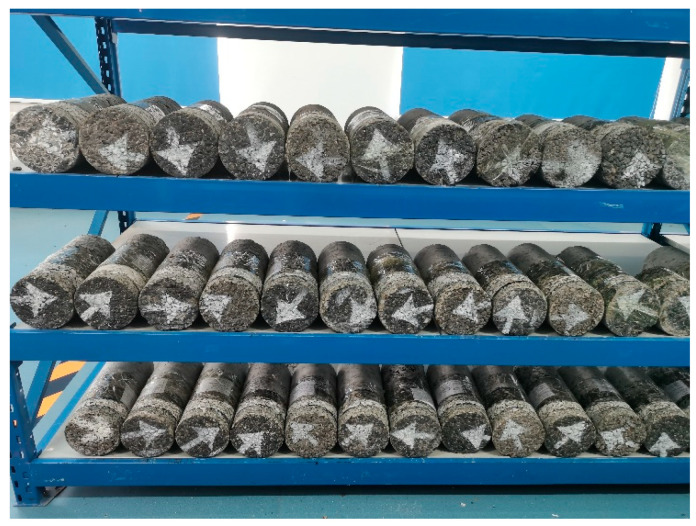
Pavement core sample.

**Figure 2 materials-15-06461-f002:**
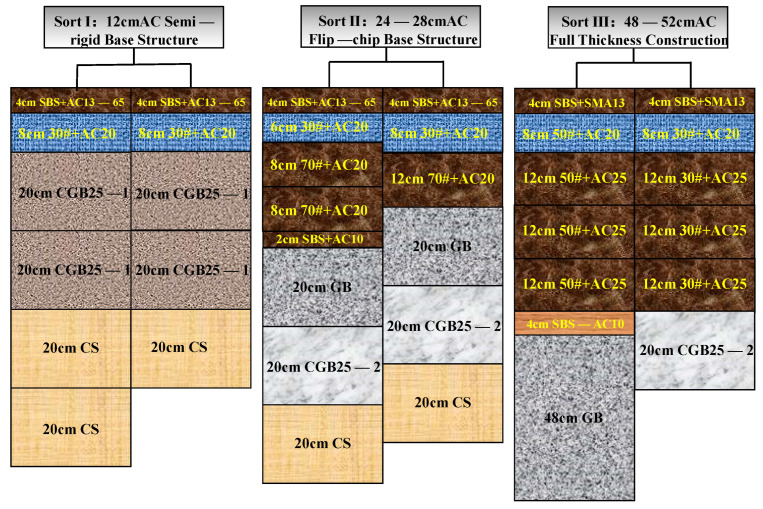
Schematic of each structure.

**Figure 3 materials-15-06461-f003:**
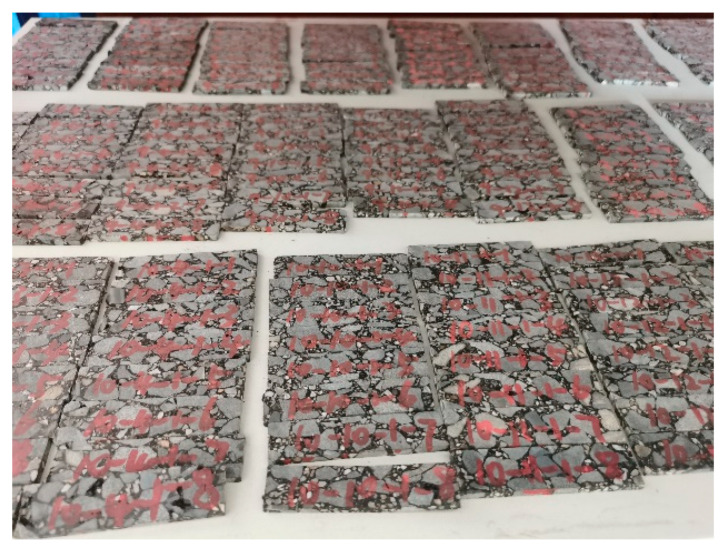
DMA specimen.

**Figure 4 materials-15-06461-f004:**
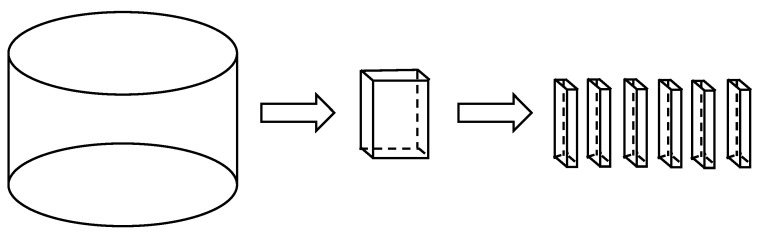
Diagram of specimen cutting.

**Figure 5 materials-15-06461-f005:**
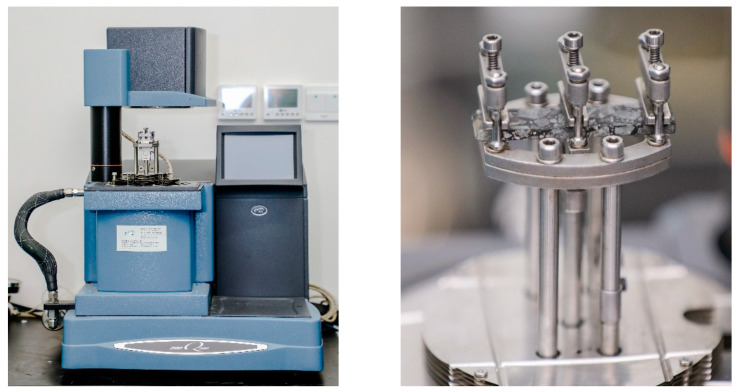
Test apparatus: DMAQ800.

**Figure 6 materials-15-06461-f006:**
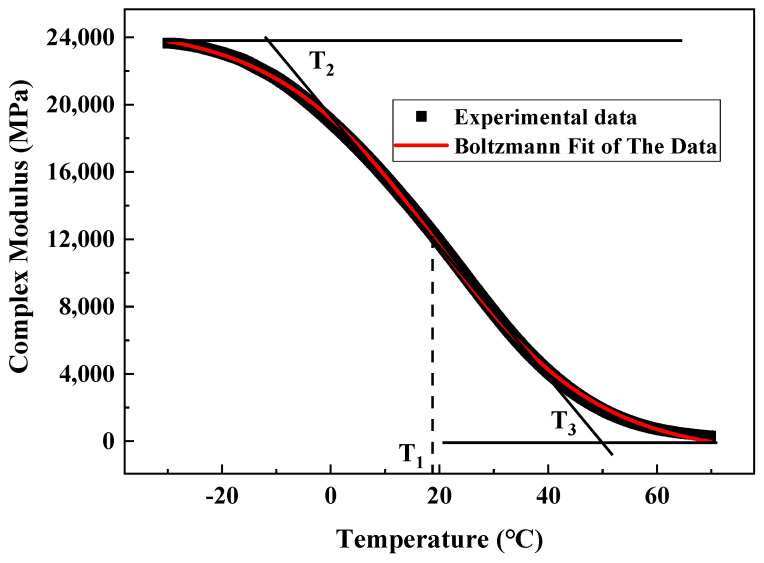
Schematic of complex modulus feature points.

**Figure 7 materials-15-06461-f007:**
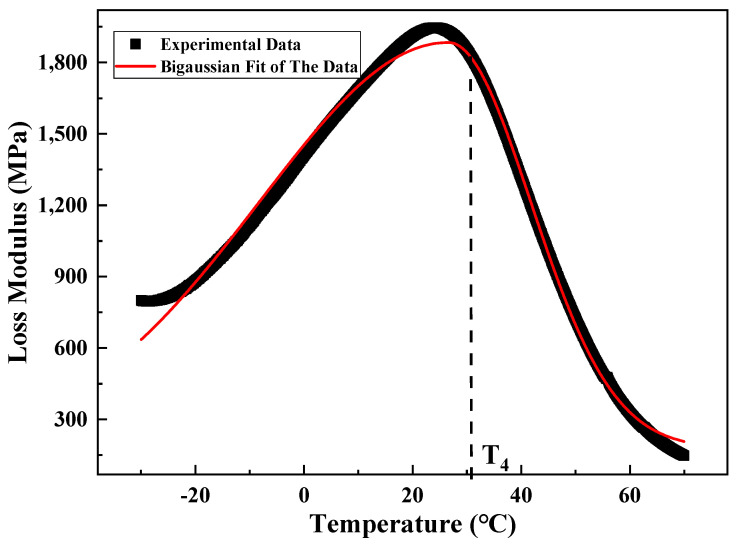
Schematic of the characteristic points of loss modulus.

**Figure 8 materials-15-06461-f008:**
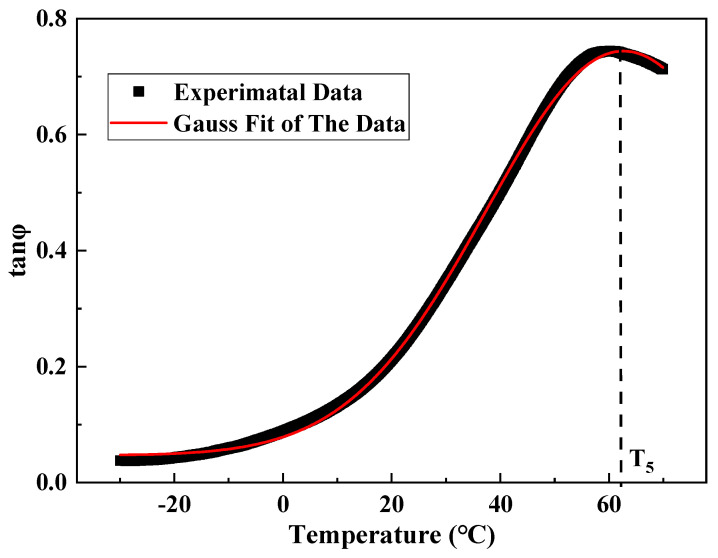
Schematic of the phase angle feature points.

**Figure 9 materials-15-06461-f009:**
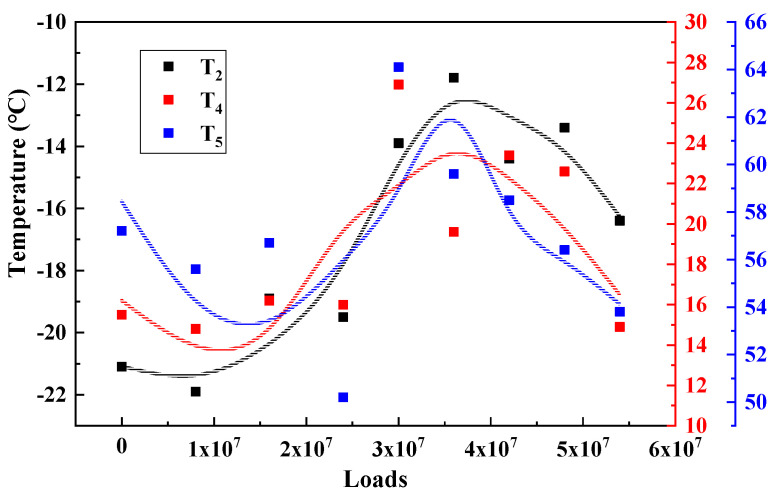
Characteristic temperature variation of STR1/3.

**Figure 10 materials-15-06461-f010:**
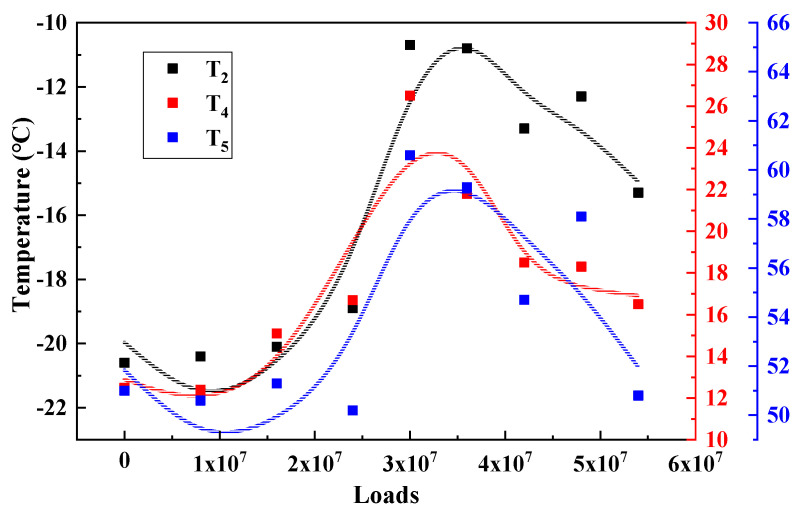
Characteristic temperature variation of STR10/12.

**Figure 11 materials-15-06461-f011:**
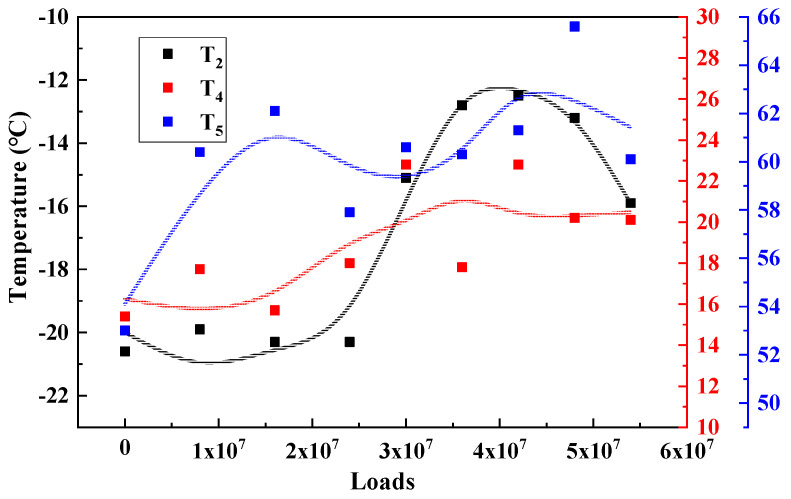
Characteristic temperature variation of STR18/19.

**Figure 12 materials-15-06461-f012:**
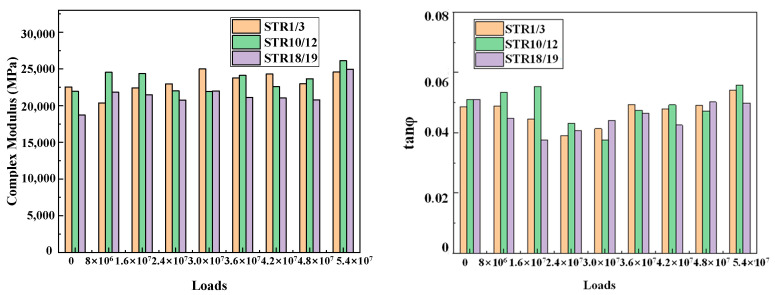
Complex modulus and phase angle at −20 °C for different loading cycles.

**Figure 13 materials-15-06461-f013:**
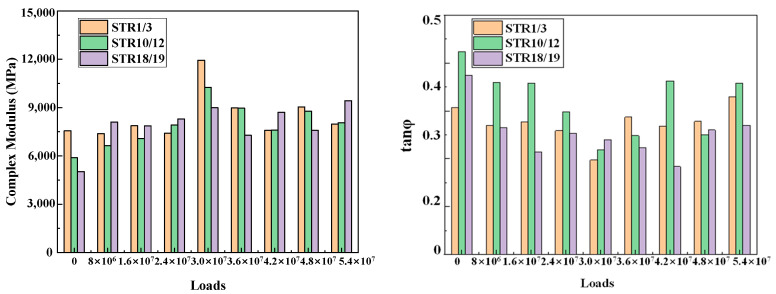
Complex modulus and phase angle at 20 °C for different loading cycles.

**Figure 14 materials-15-06461-f014:**
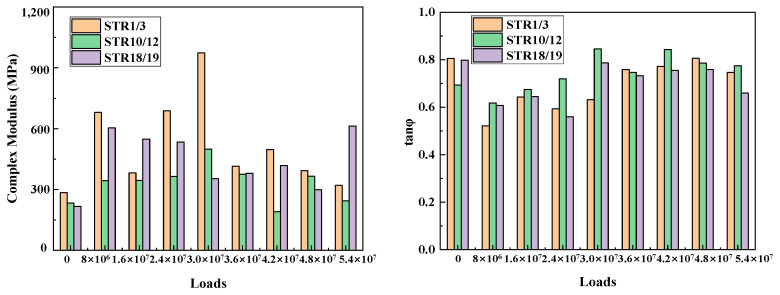
Complex modulus and phase angle at 60 °C for different loading cycles.

**Figure 15 materials-15-06461-f015:**
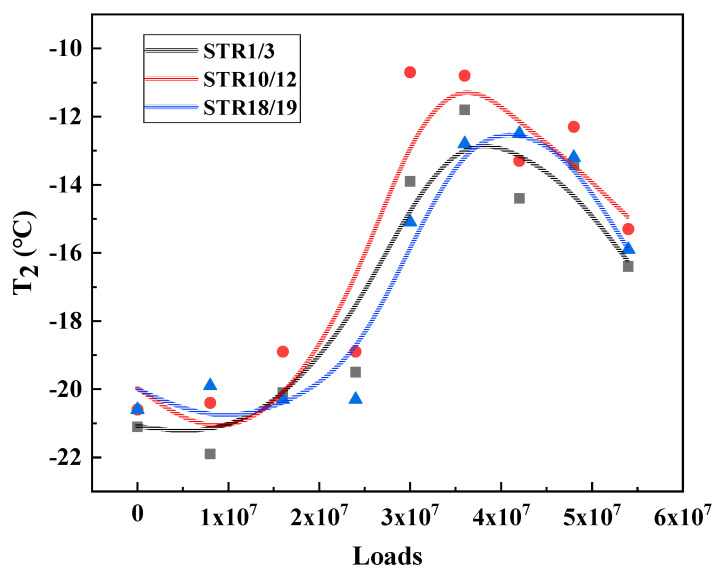
Characteristic temperature T_2_ for the three types of structures.

**Figure 16 materials-15-06461-f016:**
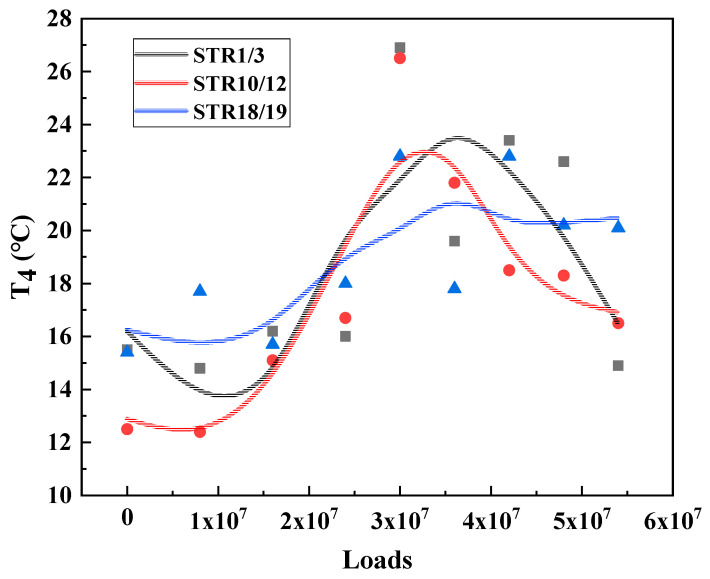
Characteristic temperature T_2_ for the three types of structures.

**Figure 17 materials-15-06461-f017:**
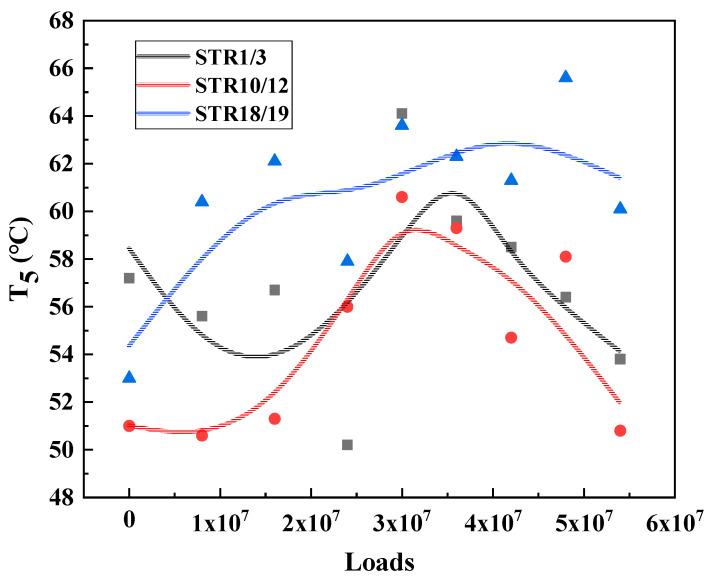
Characteristic temperature T_2_ for the three types of structures.

**Table 1 materials-15-06461-t001:** RIOHTrack coring information.

Frequency	Coring Time	Loading Axle Time	Core State
1	31 December 2018	8 million times	Carriageway (intact)
2	23 July 2019	16 million times	
3	14 January 2020	24 million times	
4	20 June 2020	30 million times	
5	3 November 2020	36 million times	
6	24 February 2021	42 million times	
7	23 June 2021	48 million times	
8	19 October 2021	54 million times	

**Table 2 materials-15-06461-t002:** Basic performance data of asphalt.

Asphalt	Ductility/cm (10 °C)	Ductility/cm (5 °C)	Penetration/0.1 mm (25 °C)	Softening Point/°C
SBS modified asphalt	48.7	28.5	63.4	72.7

**Table 3 materials-15-06461-t003:** Material gradation of the surface layer of each structure.

Structure	Material	Pass Rate of Each Grade of Sieve Hole (%)									
		16	13.2	9.5	4.75	2.36	1.18	0.6	0.3	0.15	0.075
STR1/3	AC13-65	100	98.0	60.6	34.8	25.4	17.7	13.4	9.9	8.8	7.1
STR10/12											
STR18/19	SMA13-75	100	97.7	54.7	24.9	16.7	13.9	12.5	11.4	11.0	9.7

**Table 4 materials-15-06461-t004:** Comparison of STR1/3 and STR10/12 characteristic temperature T_2_.

Loading Axle Time (Million)	STR1/3	STR10/12	Difference
8	−21.1	−20.6	−0.5
16	−21.9	−20.4	−1.5
24	−20.1	−18.9	−1.2
30	−19.5	−18.9	−0.6
36	−13.9	−10.7	−3.2
42	−11.8	−10.8	−1
48	−14.4	−13.3	−1.1
54	−13.4	−12.3	−1.1
	−16.4	−15.3	−1.1

**Table 5 materials-15-06461-t005:** Comparison of STR1/3 and STR10/12 characteristic temperature T_4_.

Loading Axle Time (Million)	STR1/3	STR10/12	Difference
0	15.5	12.5	3
800	14.8	12.4	2.4
1600	16.2	15.1	1.1
2400	16.0	16.7	−0.7
3000	26.9	26.5	0.4
3600	19.6	21.8	−2.2
4200	23.4	18.5	4.9
4800	22.6	18.3	4.3
5400	14.9	16.5	−1.6

**Table 6 materials-15-06461-t006:** Comparison of STR1/3 and STR10/12 characteristic temperature T_5_.

Loading Axle Time (Million)	STR1/3	STR10/12	Difference
0	57.2	51.0	6.2
800	55.6	50.6	5
1600	56.7	51.3	5.4
2400	56	50.2	5.8
3000	64.1	60.6	3.5
3600	59.6	59.3	0.3
4200	58.5	54.7	3.8
4800	56.4	58.1	−1.7
5400	53.8	50.8	3

## Data Availability

Not applicable.

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
