# Peer review of "Long-Term Performance Evolution of RIOHTrack Pavement Surface Layer Based on DMA Method"

_materials, 2022, doi:10.3390/ma15186461_

Round 1
Reviewer 1 Report
Dear authors,
I liked to read this paper. Was interesting and in general the DMA has some advantages. In the attached file I provided some comments and questions and included few points for small chages.

Author Response
Dear editors and reviewers,
Thank you for your careful reading, helpful comments and constructive suggestions, which have greatly improved the presentation of our manuscript. We have carefully considered all of the reviewers' comments and have revised our manuscript accordingly. Details can be seen in the revised version of the tracked changes. In the uploaded response letter, we have summarized our response to each of the reviewers' comments. We believe that our responses have addressed all of the reviewers' questions well. We hope that our revised manuscript will be accepted for publication.
Yours sincerely.
Zhimin Ma

Reviewer 2 Report
For this paper, this study conducted experimental research on the surface course asphalt mixture of three categories and six typical structures of RIOHTrack based on the Dynamic Mechanical Analysis method. Moreover, this study explored the performance evolution law of asphalt mixture under the coupling action of load and environment in the process of loading from 0 million to 54 million standard axle times. I found that the manuscript is very informative and presents the useful results of the experiment.
General comment
1. Suitable keyword for the paper.
2. Well written for the abstract.
However, I would like the authors please correct and explain some comments and questions as followings.
1. Recheck the minor English spelling errors. The language of the paper should be rechecked. There are several grammar error.
2. In order to make the paper more useful for the reader, the authors should report a comparison of the novel proppant in comparison with the ones actually used for the same process. (give more information about it). Even if, of course, this is not a review on the proppants, in order to understand why this one is important and how its characteristic make it a useful product. Introduction should be enhanced and needs to correlate between paragraph to paragraph. The application of this research output is not well highlighted. Suggest to propose and link the research output toward engineering application.
2. However, in my point of view there is some points which have to be taken in account in order to improve the work. Firstly, the Introduction has to be reviewed. It is too summarized and the sentences need to be better linked.
3. The authors should manage the order of descriptions in all Tables and Figures. Explanation of any figures and tables should be placed as close as possible.
4. Conclusions are weak and not supported by comparisons with previous studies. The Conclusion should be rewritten; as presented it is not full representative of the entire work done.
Author Response
Dear editors and reviewers.
Thank you for your careful reading, helpful comments and constructive suggestions, which have greatly improved the presentation of our manuscript. We have carefully considered all of the reviewers' comments and have revised our manuscript accordingly. Details can be seen in the revised version of the tracked changes. In the uploaded response letter, we have summarized our response to each of the reviewers' comments. We believe that our responses have addressed all of the reviewers' questions well. We hope that our revised manuscript will be accepted for publication.
Yours sincerely.
Zhimin Ma
